# OpenReview forum: "It's a TRAP! Task-Redirecting Agent Persuasion Benchmark for Web Agents"
_ICML.cc/2026/Conference — ICML 2026 regular_

### Official Review · Reviewer_Vzmz · 2026-03-07

**Soundness:** 3
**Presentation:** 3
**Significance:** 3
**Originality:** 3
**Overall Recommendation:** 4
**Confidence:** 5

**Summary:**

This paper introduces a benchmark (TRAP) for studying persuasion techniques in web agents. It’s a multi-turn agent framework with multiple environments that can be extended to new injections. Each injection is composed of five components: interaction vector, human persuasion principles, LLM manipulation methods, location of injection, and tailoring. Their results show systematic patterns of persuasion across closed and open models.

**Compliance With Llm Reviewing Policy:**

Affirmed.

**Final Justification:**

I recommend accepting this paper, but there are some limitations.

The paper is well motivated, clearly written, and well presented. It introduces a valuable framework for studying the effect of adversarial attacks on agents, which can be extended to new attacks and environments.

The thorough analysis of the results shows that the benchmark is useful to find vulnerabilities in current models, but the authors stopped there without looking at methods for mitigating these effects. Since there are other frameworks for similar purposes, I think they could've gone a bit further. However, I still think this contribution is valuable for the safe development of agents.

**Key Questions For Authors:**

1. Could you describe the process of expanding to different injections or environments? From a practical point of view, see how much work it takes.
2. Looking at Table 1, it seems that better models are more robust. For those reasoning models, could you analyze their reasoning traces? Are they aware of the adversarial attacks? What do they reason about when they are successfully attacked?

**Limitations:**

Yes

**Strengths And Weaknesses:**

### Strengths

- The paper is well motivated, clearly written, and easy to follow
- Other presentation aspects, like figures, tables, and the color coding, are very good
- As agents are promised to leak into our daily lives, it’s important to study how easy it is to influence them maliciously. This paper introduces a framework to do just that in a way that can be expanded to more injections (as newer ones come out) and more environments.
- Thorough analysis of the results (e.g. do injections discovered with a certain model transfer to others? What text injection works best? What interface injection is more effective?) showing that the benchmark is useful to find vulnerabilities in current models.

### Weaknesses

- The models are a bit out of date, so I wouldn’t say they’re frontier models. If it’s not very expensive, it’d be nice to have results from frontier reasoning models: Claude 4.6 Sonnet, Gemini 3.1 Pro, and GPT-5.4. I wonder if ASR would be even lower (even in that case, I think the framework is valuable).
- No experiments for trying to mitigate these effects, e.g., seeing if models can be steered or fine-tuned to avoid these traps.
- No analysis as to why this happens, even when reasoning traces are available.
- It would’ve been nice to have a human baseline to see what’s reasonable to expect.

---

> ### Author Rebuttal · Authors · 2026-03-31
>
> Thank you for acknowledging the importance and timeliness of the problem. We address each concern below.
>
> ---
>
> > **W1:** Newer models
>
> Thank you for this suggestion. We agree that evaluating the latest reasoning models would strengthen the paper. We were unable to complete these runs within the rebuttal window, but we commit to including results for Claude 4.6 Sonnet, Gemini 3.1 Pro, and GPT-5.4 in the camera-ready version.
> We note that our current evaluation already includes strong commercial models: GPT-5 (ASR 13%, benign utility 89%) and Claude Sonnet 3.7 (ASR 20%, benign utility 83%), both of which were frontier at the time of submission. The pattern we observe, that more robust models show lower ASR while maintaining high benign utility, is consistent with the reviewer's intuition that newer reasoning models may be even more resistant. If confirmed, this would further validate TRAP's discriminative power: the benchmark captures meaningful variation across the robustness spectrum rather than saturating at either extreme.
> TRAP's modular design makes extending to new models straightforward (no changes to the attack pipeline or environments are needed), so we anticipate no obstacles to completing these evaluations before the camera-ready deadline.
>
> ---
>
> > **W2:** Mitigation & Defenses
>
> Thank you for raising this point. Our primary objective is to expose practical vulnerabilities and provide a framework for benchmarking, rather than to evaluate defences. As in prior foundational work on adversarial attacks (e.g., Kurakin et al., ICML 2017), clearly identifying and characterising the threat is a necessary first step before rigorous defences can be designed.
> We agree that mitigation evaluation is important and are developing a follow-up study focused on defences within the TRAP framework, evaluating:
> - Real-time monitoring at agent entry/exit points using lightweight classifiers to detect injected elements
> - Cross-checking agents that review planned actions before execution
> - Sandboxed simulation pre-runs to detect hazardous deviations before deployment
> These defences will be evaluated within our framework in terms of effectiveness, computational overhead, and remaining failure modes.
>
> ---
>
> > **W3:** Reasoning traces
>
> Thank you for this suggestion. We have addressed this in Appendix E ("Agent Reasoning Traces"), with curated traces visualised at https://anonymous.4open.science/r/TRAP---ICML. The examples cover three categories:
> attack success, where agents rationalise compliance (e.g., Claude Sonnet 3.7 interpreting an injection as a legitimate terms-and-conditions gate, GPT-5 clicking with no chain-of-thought at all);
> attack resistance, including explicit detection (Claude identifying injected content as "suspicious" and reporting it to the user) and silent ignoring (GPT-5 scrolling past the same injection that fooled DeepSeek-R1); and
> edge cases such as stochastic outcomes (Gemini producing opposite results on the same injection across runs).
>
> ---
>
> > **W4:** Human Baseline
>
> We appreciate this suggestion. We expect a human baseline would be near zero, and deliberately so - TRAP's injections target agent-specific failure modes (compliance with embedded instructions, inability to distinguish legitimate UI elements from injected ones) rather than human-convincing social engineering. **A human reader would trivially recognise a fabricated button or directive planted in a product review or comment field; the benchmark's value lies in showing that frontier agents cannot.** We note this distinction explicitly in Section 3.5, where we describe the single-click metric as capturing the agent's susceptibility to instruction-following hijacks. We agree a minimal human study would make this contrast explicit, and we note it as a valuable direction for future work.
>
>
> We hope these responses address your concerns and would be grateful if you would consider updating your score accordingly.

---

> > ### Author Rebuttal · Reviewer_Vzmz · 2026-04-03
> >
> > Thank you for addressing most of my comments. I think this paper can be accepted, but I still think some weaknesses limit its impact (e.g., further evaluation such as mitigation and defenses), so I'm keeping my score.

---

### Official Review · Reviewer_ndgY · 2026-03-08

**Soundness:** 2
**Presentation:** 3
**Significance:** 2
**Originality:** 2
**Overall Recommendation:** 4
**Confidence:** 2

**Summary:**

This work introduces the Task-Redirecting Agent Persuasion (TRAP) Benchmark for studying how persuasion techniques could affect LLM agents on realistic web tasks. They found that small interface or contextual changes often doubled success rates, showing psychologically driven vulnerabilities. Specifically, button-based injections are more than three times as effective as hyperlinks. Light contextual tailoring increases ASR by up to nearly six times.

**Compliance With Llm Reviewing Policy:**

Affirmed.

**Final Justification:**

The authors addressed my concerns

**Key Questions For Authors:**

- For reasoning models, could you share some reasoning traces for patterns distinguishing successful resistance from susceptibility? I think even a small-scale analysis (e.g., whether the agent explicitly flags the content as suspicious or reasons about task relevance) would further strengthen the paper's contribution beyond aggregate ASR reporting.
- Could you at least provide error bars for Table 1?
- How is the injection length designed? It's mentioned that on average, it's 787 characters. How do you justify this as a realistic injection length, and could it be that the longer the injection, the more likely it is to succeed?
Happy to raise my score if these questions and the above-mentioned weaknesses are addressed.

**Limitations:**

Yes.

**Strengths And Weaknesses:**

Strengths:
- This benchmark is comprehensive in the sense that it covers 7 persuasion principles to study, and includes 5 LLM manipulation methods.
- This work breaks the attacks into 5 dimensions (interaction vector, persuasion principle, LLM manipulation method, injection location, and tailoring). I think this is a useful contribution because it enables systematic analysis of which specific components drive attack success.
- This work provides concrete actionable insights: buttons are 3.5× more effective than hyperlinks, light tailoring can boost ASR by up to ~6×, and the transferability analysis shows that attacks developed against the most robust model (gpt-5) transfer widely to weaker ones (82.5% average).

Weaknesses:
- Only 18 benign tasks are covered. With so few tasks, it's hard to know whether the observed ASR patterns generalize or are specific to the task formulations.
- No error bars reported throughout the paper.

---

> ### Author Rebuttal · Authors · 2026-03-31
>
> Thank you for acknowledging the importance and timeliness of our paper. We address each concern below.
>
> ---
>
> > **W1:**  Only 18 benign tasks are covered.
>
> We clarify that our task design keeps generalisability in mind. The 18 base tasks were selected as the most representative ones from REAL's web environments to cover qualitatively distinct user goals (scheduling, communication, commerce, hiring, delivery etc.) across 6 popular website clones. Through the combinatorial attack space (7 × 5), each task yields 35 qualitatively distinct attacks, producing 630 unique attack scenarios. With 6 models evaluated on all 630 (3,780 runs total), this design keeps attack generalisable with careful cross-model evaluation.
> Importantly, the observed ASR patterns are not artefacts of specific task formulations. The dominant effects (buttons outperform hyperlinks by 3.5×, tailoring increases ASR by up to ~6×) hold consistently across websites and models. The variation in persuasion principle and manipulation method drives substantially more ASR changes than variation in the base task. From a design standpoint, 18 tasks is a reasonable number, and systematically varying attack construction is more informative for understanding agent vulnerability than expanding base tasks with a fixed attack strategy.
> That said, expanding the task set could be a natural next step, and TRAP's modular infrastructure makes this straightforward for the community to put in more tasks easily.
>
> ---
>
> > **W2:** No error bars reported. & **Q2:** Could you at least provide error bars for Table 1?
>
> Thank you for flagging this. We have added 95% binomial confidence intervals to Table 1 (ASR) in the revised manuscript. The cross-model differences remain statistically robust. For example, the GPT-5 to DeepSeek-R1 gap is approximately 8× wider than the confidence intervals.
> **Table 1: Results of evaluation on benign utility and ASR.** Benign utility shows how well models complete the benign task, while ASR (n=630) shows how often they follow adversarial injections. ASR error margins are 95% binomial confidence intervals.
> | **LLM Model** | **Benign Utility** | **Attack Success Rate (ASR)** |
> |---|---|---|
> | GPT-5 | 89% | 13% ±2.6 |
> | Claude Sonnet 3.7 | 83% | 20% ±3.1 |
> | Gemini 2.5 Flash | 61% | 30% ±3.6 |
> | GPT-OSS-120B | 61% | 27% ±3.5 |
> | DeepSeek-R1 | 67% | 43% ±3.9 |
> | LLaMA 4 Maverick | 22% | 17% ±2.9 |
>
> ---
>
> > **Q1:**  Share Reasoning Traces
>
> Thank you for this suggestion. We have addressed this by adding example reasoning traces, which can be seen in the experiment logs from: https://anonymous.4open.science/r/TRAP---ICML. The reasoning traces show interesting insights on how the agents flag content as suspicious or show resistance to attacks:
> - Attack success: agents rationalise whether to comply or not (e.g., Claude Sonnet 3.7 interprets an injection as a ‘legitimate terms-and-conditions gate’ and comply);
> - Attack resistance: agent shows explicit detection of attacks (Claude identifies the injected content as "suspicious" and reports it to the user) or silent ignoring (GPT-5 scrolls past the injection and shows no resistance)
> - Stochastic outcomes (Gemini producing opposite results by either proceeding or avoiding the link on the same injection runs with different random seeds).
>
> This reasoning traces help us understand why attacks succeed/fail and strengthens our contribution beyond aggregate ASR reporting.
>
> ---
>
> > **Q3:** Injection length design.
>
> Thank you for this question. Injection length was deliberately chosen to reflect real-world content. We sampled user-generated text (reviews, comments, posts) from each of the 6 websites, yielding a tightly controlled distribution: mean 787 characters, CV ≈ 12% (SD = 98), well within typical platform limits of 3,000–5,000 characters.
> On whether longer injections succeed more: injection length variation is driven mainly by persuasion principle and manipulation method (e.g., Chain-of-Thought averages 815 characters vs. 753 for Override; Reciprocity averages 864 vs. 753 for Scarcity). A raw length–ASR correlation would therefore conflate these qualitative design dimensions. Since we analyse ASR broken down by these dimensions (Sections 4.3–4.4), the factors that drive both length and ASR are already reported (buttons vs. hyperlinks 3.5×, tailoring up to 6×, cross-model gaps 13%–43%), and these effects explain the ASR far more than injection length variation. This aligns with findings showing little correlation between ASR and injection length: Yu et al. (USENIX Security 2024) found moderate correlation (ρ = 0.26) between length and jailbreak success, and Andriushchenko et al. (ICLR 2025) achieved 100% ASR with just 10-character injections.
>
> ---
>
> We hope these revisions: clarifying our design choices on task design and injection length, and adding reasoning traces and confidence intervals, address the concerns you raised. If you agree, we would be grateful if you would consider updating your score.

---

> > ### Author Rebuttal · Reviewer_ndgY · 2026-03-31
> >
> > Thank you for the thorough rebuttal. I appreciate the effort in addressing my concerns.
> >
> > I am happy to raise my score accordingly.

---

### Official Review · Reviewer_z3MB · 2026-03-12

**Soundness:** 3
**Presentation:** 2
**Significance:** 2
**Originality:** 3
**Overall Recommendation:** 4
**Confidence:** 4

**Summary:**

This paper introduces TRAP, a benchmark for evaluating the adversarial robustness of web agents against prompt injection attacks. The benchmark is grounded in six cloned real-world platforms and decomposes attacks into modular components, enabling systematic analysis of injection strategies and their effectiveness.

**Compliance With Llm Reviewing Policy:**

Affirmed.

**Final Justification:**

The rebuttal improves empirical support and clarifies the design. Across both the paper and rebuttal, there are minor editing issues (e.g., near-duplicate paragraphs), suggesting the need for more careful proofreading. I raise my score, but maintain my original concerns about the paper’s overall contribution.

**Key Questions For Authors:**

no further questions.

**Limitations:**

yes discussed

**Strengths And Weaknesses:**

**Strengths**

Focus on a critical and timely challenge in web agent security, with attacks from real-world platforms.

The five-component decomposition of prompt injections is well-motivated and effectively illustrated through figures.

The main text is visually accessible, with rich figures aiding comprehension of the framework. However, this comes at the cost of space for deeper empirical analysis.

Evaluations are broad in scope, covering prompt transferability and attribute-level analysis covering most of the injection components.

**Weakness**

Some empirical claims lack supporting evidence. For instance, the choice of AXTree over screenshot/DOM representations is justified with: "We evaluated agents using all three types, both individually and in combination, and noticed very small differences in benign utility and attack success rates." If these experiments were conducted, even a small-scale ablation in the appendix would substantively support this claim. Alternatively, citing prior literature that establishes this finding would be more appropriate.

The text requires additional editing. Lines 411–422 contain two paragraphs with substantially overlapping content:
"For LLaMA-4 Maverick, nearly all successful hijacks (both targeted and non-targeted) originate from the About section, further reinforcing that..."
"For LLaMA-4 Maverick, almost all successful prompt injections (both targeted and non-targeted) originate from the About section, further indicating that weaknesses..."

The claim "We release a modular, extensible framework" is not well-supported. It seems that there's no code, project page, or supplementary file appears to be provided. If a release is intended, the relevant link should be included; otherwise, "propose" could be a more appropriate than "release."

While the modular decomposition is desirable for controlled evaluation (Sections 4.3–4.6), the gap between this abstraction and injections  in the wild is a meaningful limitation that is not explicitly discussed.

Figures are better to be provided in vector format. As currently included, fine-grained text in figures is difficult to read.

---

> ### Author Rebuttal · Authors · 2026-03-31
>
> Thank you for acknowledging the importance and timeliness of the problem. We address each concern below.
>
> > **W1:** Unsupported empirical claims (e.g., AXTree vs. screenshot/DOM ablation).
>
> We thank the reviewer for this suggestion. We added a direct ablation in Table 14 (Appendix). We excluded full-page HTML early due to prohibitive token cost and no observed benefit, leaving AXTree and screenshots as viable modalities. On GoCalendar with Gemini 2.5 Flash (n≈105 per condition), AXTree-only yielded 8.7% ASR and 12.5% benign utility, while AXTree+Screenshot yielded 10.5% ASR and 8.6% benign utility - differences that are not statistically significant (p>0.5, Fisher's exact test). This, combined with AXTree's lower cost, compatibility with non-VLMs, and higher reproducibility, led us to use AXTree-only in the main experiments.
>
> **Table 14.** Observation modality comparison on GoCalendar (Gemini 2.5 Flash). Adding screenshots to AXTree observations does not significantly change ASR or benign utility (*p*>0.5, Fisher's exact test for both).
> | Observation | *n* | ASR | Benign Utility |
> |---|---|---|---|
> | AXTree only | 104 | 8.7% (9) | 12.5% (13) |
> | AXTree + Screenshot | 105 | 10.5% (11) | 8.6% (9) |
>
>
> ---
>
>
> > **W2:** Overlapping paragraphs (Lines 411–422)
>
> Thank you for catching this. The two paragraphs have been merged into a single, non-redundant sentence: *"For LLaMA-4 Maverick, nearly all successful prompt injections (both targeted and non-targeted) originate from the About section, reinforcing that vulnerabilities cluster around semantically rich profile text rather than task-neutral areas."*
>
>
> ---
>
>
> > **W3:** Released framework
>
> Thank you for flagging this. We now provide an anonymous repository containing the full framework (code, injection templates, and evaluation pipeline) at https://anonymous.4open.science/r/trap-CC4B/README.md. We have updated the manuscript to include this reference. We hope this addresses the concern directly.
>
>
> ---
>
>
> > **W4:** Gap between modular decomposition and real-world injections
>
> We agree that the gap between modular and naturalistic injections is worth discussing, and we have added a sentence to the Limitations paragraph acknowledging this. However, we note that the modular decomposition is a deliberate design choice: isolating individual components (persuasion principle, interface type, location, tailoring) is what enables the controlled comparisons and derives valuable insights in Sections 4.3-4.6. Importantly, the end-to-end evaluations in Table 1 use fully composed injections that combine multiple components, providing a complementary view closer to real-world conditions. The added limitation sentence now reads: "The modular decomposition enables controlled evaluation but abstracts away the organic co-occurrence of components found in real-world injections; future work could supplement these findings with naturalistic attack compositions."
>
>
> ---
>
>
> > **W5:** Figures should be in vector format
>
> We agree that vector figures can improve readability. We will replace all figures with PDF vector exports in the camera-ready submission. The current PNG figures are placeholders from the development pipeline; the final versions will be generated directly as `.pdf` from the plotting scripts, ensuring fine-grained text and lines remain crisp at any zoom level.
>
>
> ---
>
>
> We hope these revisions address all the concerns raised in your review. If you agree, we would be grateful if you would consider updating your score to reflect these changes.

---

> > ### Author Rebuttal · Reviewer_z3MB · 2026-04-03
> >
> > Thank you for the rebuttal. I appreciate the additional clarifications and the newly provided experimental details (e.g., AXTree ablation and repository link), which help address several of my concerns regarding empirical support.
> >
> > However, I have some reservations.
> > I still find that the gap between the modular abstraction and real-world attack scenarios limits the paper’s impact. Although the added experiments are helpful, some of these aspects would benefit from stronger integration into the main paper rather than being deferred to appendices or lightly presented.
> >
> > And there appears to be a minor inconsistency in the rebuttal: the authors refer to “Table 14 (Appendix)” for the new ablation, but Table 14 already exists in the original submission. It is unclear whether this refers to a newly added table (e.g., Table 15) or a renumbering.
> >
> > Overall, I appreciate the effort in addressing the feedback, but I prefer to maintain my original score.

---

> > > ### Author Response · Authors · 2026-04-06
> > >
> > > Thank you for the follow-up and for noting that the AXTree ablation and repository link address the empirical-support concerns. We focus our response on the two points that remain: main-paper integration of the modular-vs-naturalistic discussion, and the framing of the abstraction gap itself.
> > >
> > > ---
> > >
> > > **Strengthening main-paper integration.** We agree this should not sit in a single Limitations sentence or be inferred from Table 1. In the revised manuscript, we added an explicit paragraph to Section 1 that names the abstraction-versus-naturalistic-injection gap directly, positions TRAP as a controlled, reproducible estimate of agent vulnerability whose attack components are anchored to documented real-world incidents (which we have previously cited on page 1), and acknowledges that in-the-wild attacks may differ in adaptivity, composition, and context. We also moved the modality ablation from the appendix into Section 3, alongside the AXTree justification. These changes make the gap a load-bearing part of the framing.
> > >
> > > ---
> > >
> > > **On the framing of the gap itself.** We would like to engage with this once more, as we believe "modular abstraction versus real-world attacks" may not fully reflect what TRAP evaluates. The 35 templates are already composed attacks, not isolated components. Each combines a persuasion principle, an LLM manipulation method, an interface vector, a location, and tailoring into a single realised injection. An agent encountering any TRAP scenario faces a fully composed attack, not a stripped-down single-factor probe. Sections 4.3 to 4.6 decompose these composed attacks post hoc to isolate which design choices drive success (the controlled view). Table 1 reports end-to-end performance on the composed attacks themselves (the naturalistic view). Both are evaluated on the same 630 scenarios.
> > >
> > > Controllability and naturalism are a standard benchmark trade-off. AgentDojo [1], InjecAgent [2], and ASB [3] all construct injection attacks from templated components (canonical attack types combined with user tasks or tool responses via placeholder substitution) rather than scraping organic in-the-wild injections, because reproducibility and controlled variation are prerequisites for measuring which factors matter. TRAP's contribution within this tradition is that its factorial structure exposes sensitivities (3.5x for button versus hyperlink, up to 5.6x for tailoring) that monolithic benchmarks cannot detect, giving defenders concrete targets for mitigation design.
> > >
> > > ---
> > >
> > > **Table numbering.** Apologies for the confusion: the new modality ablation is Table 15 in the revised manuscript, not a replacement of the existing Table 14. ICML 2026 policy does not permit attaching revised manuscripts during rebuttal, so these changes will become visible in the camera-ready.
> > >
> > > ---
> > >
> > > We hope these clarifications and the planned integration address the remaining concern.
> > >
> > > ---
> > >
> > > [1] E. Debenedetti, J. Zhang, M. Balunović, L. Beurer-Kellner, M. Fischer, F. Tramèr. AgentDojo: A Dynamic Environment to Evaluate Prompt Injection Attacks and Defenses for LLM Agents. Advances in Neural Information Processing Systems (NeurIPS), 2024.
> > >
> > > [2] Q. Zhan, Z. Liang, Z. Ying, D. Kang. InjecAgent: Benchmarking Indirect Prompt Injections in Tool-Integrated Large Language Model Agents. Proceedings of the Association for Computational Linguistics (ACL), 2024.
> > >
> > > [3] H. Zhang, J. Huang, K. Mei, Y. Yao, Z. Wang, C. Zhan, H. Wang, Y. Zhang. Agent Security Bench (ASB): Formalizing and Benchmarking Attacks and Defenses in LLM-based Agents. International Conference on Learning Representations (ICLR), 2025.

---

### Official Review · Reviewer_HxSy · 2026-03-17

**Soundness:** 3
**Presentation:** 3
**Significance:** 3
**Originality:** 2
**Overall Recommendation:** 4
**Confidence:** 4

**Summary:**

This paper introduces TRAP, a benchmark for evaluating persuasion-driven prompt injection attacks against LLM-based web agents. The benchmark is built on cloned versions of six real-world websites and constructs attacks through a modular design combining interface elements, persuasion principles, LLM manipulation techniques, injection locations, and contextual tailoring. The authors evaluate six frontier models and show that agents remain vulnerable to prompt injections, with an average attack success rate of about 25%. The study further analyzes which persuasion strategies, manipulation methods, and interface designs most effectively redirect agents.

**Compliance With Llm Reviewing Policy:**

Affirmed.

**Final Justification:**

The rebuttal addressed several of my concerns.

**Key Questions For Authors:**

See weaknesses above

**Limitations:**

- The benchmark focuses on text-based injections and does not evaluate multimodal attacks (e.g., image-based prompt injections).
- The work studies attack measurement only, leaving defensive strategies and mitigation methods for future work.

**Strengths And Weaknesses:**

Strengths

- Addresses an important and timely problem: prompt injection vulnerabilities in LLM-based web agents.
- Evaluates multiple modern models across realistic website environments, providing useful empirical insights into agent vulnerabilities.
- The analysis of factors such as interface design (buttons vs hyperlinks) and cross-model transferability is interesting and potentially useful for future research.

Weaknesses

- Limited novelty in attack techniques. Most manipulation methods (e.g., CoT injection, adversarial suffixes) are already well studied; the main contribution is the benchmark composition rather than new attack mechanisms or new data construction methods.
- Single-step success metric. Defining success solely as clicking the injected element may oversimplify real attack scenarios and ignores downstream agent behavior.
- No proposed defenses. The paper focuses entirely on measuring vulnerabilities without exploring mitigation strategies or defensive baselines.

---

> ### Author Rebuttal · Authors · 2026-03-31
>
> Thank you for acknowledging the importance and timeliness of the problem. We address each concern below.
>
> ---
>
> > **W1:** Limited novelty in attack techniques.
>
> Thank you - this is an important distinction to clarify. TRAP uses well-studied attack techniques (CoT injection, adversarial suffixes, etc.) as only one attack dimension called 'manipulation method', but our contribution is a five-dimensional modular attack space (persuasion principle × manipulation method × interface type × injection location × tailoring), where all the rest of four dimensions are all novel, specific to web agents, and often omitted by the community. With this modular design, we can isolate why attacks succeed rather than merely whether they do. This controlled design of TRAP indeed reveals novel insights: Interface design (buttons vs. hyperlinks) and contextual tailoring each independently double ASR, findings that only emerge because the attack space is modularised. Prior popular benchmarks such as AgentDojo use monolithic injections without this granularity, and InjecAgent varies attacks only in a binary based setting. This level of fine-grained controlled analysis on attacks is novel and a more realistic attack ground for the community to prioritise defences.
>
> ---
>
> > **W2:** Single-step success metric
>
> Defining success solely as clicking the injected element may oversimplify real attack scenarios and ignores downstream agent behavior.
> We agree that real hijacks often involve multiple steps, and address this both in the paper (Appendix C.4, Figure 15) and below.
> The single-click metric is a deliberate method choice. It isolates the critical security boundary: the moment an agent follows a malicious instruction and transfers execution to attacker-controlled context. **Once the agent navigates to an attacker-controlled page, arbitrary follow-up attacks become possible - the attacker now controls the full observation space and can chain further injections without the constraints of embedding payloads in small user-editable fields on a legitimate site.** This framing aligns with Foerster et al. (2026), who independently adopt one-step attacks as their primary evaluation unit, arguing that multi-step variants amount to chaining instances of this initial misdirection.
> We prefer single-click over downstream harm for two reasons. First, it separates susceptibility to the injection from behaviour on the attacker's page - distinct failure modes requiring different analyses. Second, it yields an objective, reproducible metric independent of attacker creativity in designing follow-up payloads. Multi-step criteria (e.g., InjecAgent, AgentDojo) conflate these, obscuring the fine-grained attribution across attack dimensions that TRAP is designed to support.
> We also show the single click is not benign: this video https://anonymous.4open.science/r/TRAP---ICML demonstrates a complete attack chain in which the agent, after clicking an injected link, is redirected to an attacker-controlled page and induced to leak sensitive user data.
>
>
> ---
>
> > **W3:** Defences
>
> Thank you for raising this point. Our primary objective is to expose practical vulnerabilities and provide a framework for benchmarking, rather than to evaluate defences. As in prior foundational work on adversarial attacks (e.g., Kurakin et al., ICML 2017), clearly identifying and characterising the threat is a necessary first step before rigorous defences can be designed.
> We agree that mitigation evaluation is important and are developing a follow-up paper focused on defences, evaluated within our framework in terms of effectiveness, computational overhead, and remaining failure modes.
>
> ---
>
> > **L1:** Benchmark is not multimodal
>
> Thank you for raising this point. We focus on text-based injections (buttons and hyperlinks) for three reasons.
> - They represent the most realistic attack surface: on real multi-user websites, adversaries control only user-editable content and cannot introduce pop-ups, banners, modals, or custom scripts.
> - Many richer UI elements are structurally equivalent - from the agent's perspective, pop-ups and banners reduce to a clickable redirect element, making buttons a canonical representation of these redirection mechanisms.
> - Image-based injections are supported within TRAP's architecture but not yet included at scale: there is currently no scalable method for generating adversarial images for agents (to our knowledge, Aichberger et al., NeurIPS 2025 is the most recent work), and image evaluation incurs significantly higher computational cost. We are actively extending TRAP to support text-in-image attacks within user-editable areas (e.g., images in the Airbnb clone).
>
>
> ---
>
> We hope these responses address your concerns and would be grateful if you would consider updating your score accordingly.

---

> > ### Author Rebuttal · Reviewer_HxSy · 2026-04-05
> >
> > I thank the authors for their detailed and thoughtful rebuttal. The clarifications regarding the modular attack design, the rationale for the single-step success metric, and the additional experimental details (e.g., ablations and released repository) help address several of my concerns. I will raise my score.

---

### Decision · Program_Chairs · 2026-04-30

**Decision:**

Accept (regular)

**Comment:**

The paper introduced a new benchmark to evaluate how persuasion techniques mess with the performance of autonomous web agents. All reviewers agree that the problem the benchmark is addressing is timely, and the benchmark is reasonably thorough. The primary concerns are that the benchmark doesn't actually reflect real-world attack scenarios, the attack techniques are not novel, and no defenses are provided.

However, given that the authors plan to release their benchmark for others to experiment with and improve upon, I think its value to the ICML community warrants publication, even without more rigorous experiments within the paper itself.